# Food Insecurity and Socioeconomic Disadvantage in Australia

**DOI:** 10.3390/ijerph17020559

**Published:** 2020-01-15

**Authors:** Ami N. Seivwright, Zoe Callis, Paul Flatau

**Affiliations:** Centre for Social Impact UWA Business School, The University of Western Australia, Crawley 6009, Australiapaul.flatau@uwa.edu.au (P.F.)

**Keywords:** food insecurity, socioeconomic disadvantage, mental health, physical health, service use, public policy

## Abstract

Research on food insecurity in Australia has typically relied on a single-item measure and finds that approximately 5% of the population experiences food insecurity. This research also finds that demographic characteristics such as household composition and marital status affect levels of food insecurity, independent of income level. The present study examines the prevalence and correlates of food insecurity in a cohort (n = 400) of people experiencing entrenched disadvantage in Perth, Western Australia. Using the US Department of Agriculture Household Food Security Survey Module, we find that food insecurity at the household, adult, and child level is at sharply elevated levels, with 82.8% of the sample reporting household food insecurity, 80.8% and 58.3% experiencing food insecurity among adults and children, respectively. Demographic characteristics do not significantly affect levels of food insecurity, and food insecurity is associated with negative physical and mental health outcomes. Food insecurity is positively correlated with access to food emergency relief services, indicating that these services are being used by those most in need, but do not address the root causes of food insecurity. Policy and practice should focus on increasing stable access to adequate quantities and quality of food and addressing the structural causes of food insecurity.

## 1. Introduction

Food security occurs when people ‘can get enough food to eat that is safe, that they like to eat, and that helps them to be healthy. They must be able to get this food in ways that make them feel good about themselves and their families’ [1]. Food security comprises four aspects: the availability of food, the physical and financial resources to access food, the ability to utilize food, requiring safe food and water as well as the ability to safely prepare, cook and store food, and finally, the stability of food supply [2]. Definitions of food security highlight the fact that food security is not simply the absence of food scarcity and hunger, but also relates to broader issues of a healthy and balanced diet and, for this reason, food security is a prominent policy and public health issue in high-income countries [3]. 

The United States has been regularly measuring hunger at the national level since the 1960s, with detailed information about food insecurity collected since the late 1980s, and further methodological improvements occurring in the 1990s with the development of the United States Department of Agriculture (USDA) Household Food Security Survey Module (HFSSM; originally the Core Food Security Module) [4,5]. Canada has used the HFSSM in its nationally representative Community Health Survey since 2004 [6]. New Zealand has used items suited to its cultural context that thematically overlap with the HFSSM, capturing the reduction of food intake and/or substitution into lower cost food, access of food banks or other relief services, and anxiety, associated with food insecurity [5,7]. Though varying in line with broader economic conditions [8], food insecurity is found among 14% of the US population, 15% of NZ, and 12.3% of Canada [9]. 

In Australia, food insecurity has overwhelmingly been measured using a single item: ‘In the last 12 months was there any time you have run out of food and not been able to purchase more?’ In a systematic review of 57 articles on food security in Australia, McKay, Haines, and Dunn [10] found that, of the 36 studies that directly measured food insecurity, 22 (61%) used the single item measure of food security. In population-representative surveys, the single item consistently estimates the prevalence of food insecurity among the Australian population to be approximately 5% [3,11,12]. Relative to multi-item measures, single item measures of food security underreport rates of food insecurity [13]. However, even the arguably conservative estimate of 5% of the population translates to 1 million Australians affected by food insecurity. Therefore, the prevalence, predictors, and impacts of food insecurity in this context remain important and relevant avenues for research.

Prevalence of food insecurity varies by substrata of the population of high-income countries, such as ethnic minorities, the homeless, tertiary students, single parents and refugees [14,15,16]. In the Australian context, a study of Victorian university students using the FSSM found that 18% had low and 30% had very low food security [17], while a study in Queensland found that 12.7% of a university student sample were food insecure using the single item measure, and 71.8% were food insecure using the FSSM [18]. Studies of refugees using the single item measure find that between 71% [19] and 90% [20] were food insecure. Studies of older Australians, also using the single item, find lower prevalence than in the general population at 2% of Australians aged over 65 years [21] and 3% of Australians aged over 55 years [22].

In terms of predictors, while low income is the strongest and most consistent predictor of food insecurity, higher income is not a proxy for food security as income level does not always reflect the economic conditions of the household [9]. For example, households can experience significant, unexpected changes to their economic circumstances, such as the loss of an income or large household bills, which have lagged effects on household income and/or require temporary reallocation of financial resources, resulting in food insecurity. On the other hand, income does not necessarily reflect assets held, nor levels of access to other supports that prevent food insecurity. As such, while the relationship between low income and food insecurity is strong, the factors that lead to food insecurity in developed countries, particularly those with strong social safety nets, are more complex [23]. Lower education levels, single-parent household composition, unemployment, and social isolation are associated with higher food insecurity in Australia, the UK, and the US [4,12,13,17,23,24]. 

Food insecurity has significant short- and long-term impacts on physical health and social and economic participation [3]. Food insecurity often leads to stress, cycles of fasting and bingeing, and the substitution of relatively higher cost, higher nutrition food with lower cost, higher energy food, resulting in nutrient deficiencies, metabolic changes, weight loss or, seemingly paradoxically, overweight and obesity [13]. Children in food insecure households in Australia were more likely to miss days of school, miss out on school activities, and more likely to have emotional and behavioral issues [25]. Eighty percent of food insecure university students reported that their studies were negatively affected by their food insecurity. They were also three times more likely to have deferred study due to financial difficulties and twice as likely to report poor or fair health [26]. Food insecure adults are more likely to have lower self-assessed health status, higher prevalence of chronic disease such as diabetes, cardiovascular disease and depression, and lower rates of social and economic participation [13]. Elderly food insecure people were six times more likely to rate their lifestyle as unhealthy than healthy, five times less likely to have private health insurance, and four times more likely to report needing household help [21]. Therefore, although there is an inverse relationship between food insecurity and age, food insecurity exists in populations at all stages of the life course and results in significant health, social and economic consequences. 

The present study examines food security among a cohort of non-government service users assessed by service users as experiencing hardship and entrenched disadvantage in the city of Perth, Australia (population two million people). Entrenched disadvantage is not just characterized by low income. It comprises long-term, often intergenerational exclusion from social and economic opportunities, including higher education, employment, and positive social and community based relationships, that affects all domains of wellbeing [27]. Previous research reveals that, in addition to income, demographic characteristics such as household composition, educational attainment, country of birth, and employment status significantly predict food insecurity. Acknowledging this previous research and the low-income nature of our sample, we explore whether there are differences in levels of food security between groups with different demographic characteristics. Further, given that the sample use community services in one way or another, we examine the relationship between people’s access of food-related services and food insecurity. Finally, this paper examines social, health and economic outcomes related to food insecurity. 

This paper contributes to the food security literature by adding to the evidence base on the prevalence of food insecurity in high-income countries using a multi-item, validated tool. Further, it sheds light on some of the nuances of the relationship between income and food security by examining differences in levels of food security between groups within a low-income sample. Additionally, effectiveness of responses to food insecurity is explored through analysis of the relationship between service use and food insecurity. Finally, acknowledging the different impact of food insecurity in high-income nations relative to the famine and chronic undernourishment that characterize food insecurity in countries with high rates of extreme poverty [28] and the seemingly contradictory outcomes that food insecurity can bring about, such as underweight and obesity [13]; this paper contributes to the literature on the impacts of food insecurity by exploring the relationships between food insecurity and health, mental health and quality of life outcomes.

## 2. Method

Data were collected as part of a collaborative research, policy and practice project between non-government service providers and university researchers. The non-government service providers covered a broad range of service delivery domains including financial counselling programs, community mental health services, homelessness and housing support services, out-of-home care services and emergency relief services. Participants in the study were 400 family representatives that were identified by partner non-government service delivery agencies as having two or more of the following ‘eligibility criteria’ for hardship and entrenched disadvantage: reliance on welfare payments, unstable housing, unemployment or underemployment, physical or mental disability, or mental health issues, inadequate social support, and low education. These criteria were selected as known correlates of entrenched disadvantage that could be ascertained by service workers with relatively minimal burden on the worker or the potential participant. The study was conducted according to the guidelines laid down in the Declaration of Helsinki and all procedures involving research study participants were approved by The University of Western Australia Human Research Ethics Committee (Ref: RA/4/20/4793). Written informed consent was obtained from all subjects.

Participants referred by service delivery agencies that were interested in participating in the study presented at their most conveniently located agency. An interviewer on the research team explained the study in full, provided each participant with a Participant Information Form for their records, and sought informed consent. Consenting participants then completed a survey on the Qualtrics survey software platform, guided by the interviewer. The survey was approximately one hour in length and covered several domains of socioeconomic wellbeing. 

Survey data were analyzed using SPSS version 25.0 (IBM Corp. Armonk, NY, USA). Relationships between food insecurity and demographic characteristics were assessed using Pearson chi-square tests. Distributions for household size and service use were not sufficiently normal for parametric analysis based on Curran, West and Finch’s guidelines [29]. Accordingly, Spearman correlations between food security and household size and service use were performed. To examine the relationships between mental health, quality of life, and health and food security, a series of Pearson correlations were conducted. *p*-values ≤ 0.05 were considered statistically significant for all analyses.

## 3. Measures

Socio-demographic variables collected include date of birth, sex, Aboriginal or Torres Strait Islander identification, country of birth, marital status, employment status and educational attainment.

Food security was assessed using a modified version of the 18 item United States Department of Agriculture (USDA) Food Security Survey Module (FSSM). Questions pertaining to child food security were asked to participants that indicated they had a child in their household or in their care. The FSSM produces raw scores of food insecurity at the household level, among adults, and among children. Scores are then used to categorize respondents as having high, marginal, low or very low food security at the household-level and among adults, and high or marginal, low, and very low food security among children. In the FSSM version used in this study, the word “household” was replaced with “family unit”. This is because the study sought to examine entrenched disadvantage beyond the boundary of the household to account for the fact that very low-income individuals may not in fact have a dwelling, or may have a non-traditional family structure in which resources are shared across households. Consequently, we amended the standard approach and refer to food security in the family unit, in order to account for variations between household and family structures and so that scores for those experiencing homelessness include people that they are living with (perhaps sleeping rough with), and are comparable to those in stable housing. 

In addition to demographic characteristics, we examine the relationship between food security and household size; food emergency relief related-service use; and mental health, wellbeing and physical health outcomes. In relation to household size, acknowledging that self-identified family structures are not restricted to those living in the same dwelling and those usually in the household may not always represent the full set of family members that one shares food with; we asked participants to identify those people that they considered part of their family unit and, for each person, how many nights that person spent in their dwelling. The sum of nights spent in the dwelling across all members of the family unit, including the participant, was calculated to create ‘total person nights’, the sum of all of the nights that people stay with the participant per week. The total person nights were then divided by the number of days in the week to derive the ‘mean person nights’—the average number of people staying in the participant’s dwelling per night. We also determine household size using the standard approach by adults and children that usually stayed in the participant’s dwelling.

Service use was identified by asking participants to indicate whether they had accessed non-government services across a range of different categories, for example, food emergency relief, mental health services, financial counselling services, emergency accommodation, in the 12 months prior to survey. For each type of service accessed, participants were asked to identify the number of different services that they accessed. For the purposes of the current study, only food emergency relief related-service use is reported.

We use a number of measures of mental health and wellbeing and physical health. Loneliness was measured using the UCLA 3 item loneliness scale [30]. This is an indirect measure of loneliness, asking participants how often they feel they lack companionship, left out, and isolated from others. Depression, anxiety and stress were measured using the Depression Anxiety Stress Scale (DASS-21) [31]. The World Health Organization (WHO)-5 Wellbeing Index [32] was also used as a measure of current mental wellbeing. The WHO Quality of Life—Brief (WHOQOL-BREF) [33], measuring wellbeing in the domains of physical health, psychological, social relationships, and environment, was administered. Physical health was assessed through participants’ self-reporting of receiving a diagnosis from a medical professional of long-term health conditions that are considered chronic in nationally representative studies and reports [34]. 

## 4. Results

### 4.1. Sample Sociodemographic Characteristics

Key sociodemographic characteristics of the sample (n = 400) are presented in Table 1. Reflecting elevated rates of disadvantage among certain groups we find that the cohort of people surveyed are more likely than the general Australian population to be Indigenous, not employed, sole parent women, homeless, and living in public or community rental housing. 

### 4.2. Prevalence of Food Insecurity

Table 2 outlines the proportion of the sample in each category of food insecurity for three of the Food Security Scales that can be calculated from the FSSM: Household (family unit), Adult, and Child. The Household Food Scale is calculated using all 18 items of the FSSM for households with children, and 10 items for those without children. With regard to food security at the household level and among adults, the majority (59.3% and 62.0%, respectively) of the sample experienced very low food security in the 12 months prior to survey. Food security was slightly higher among children, with a larger proportion (41.7%) experiencing high or marginal food security (note that the USDA FSSM labels high or marginal food security among children as a single category) than the proportion (11.1%) experiencing very low food insecurity. 

### 4.3. Differences in Food Security by Demographic Characteristics

Chi-square tests of differences in food security between groups of different demographic characteristics were conducted. Food security is assessed using the FSSM either in categorical form (high, marginal, low or very low food security) or in binary form; scores in the high or marginal categories are categorized as food secure while scores in the low and very low categories as food insecure. For both categorical and binary food security among adults and among children, there were no statistically significant differences between males and females, Indigenous and non-Indigenous participants, between those of different marital statuses, nor between different household compositions. There were also no statistically significant differences in food security between the employed, unemployed, and those not in the labor force, nor were there significant differences in food insecurity at the household level or among adults between those that were solely welfare dependent and those that had some form of wage or salary-based income. Among adults, those that did not complete high school were more likely than those that did complete high school to report very low food security (68.2% and 57.3%, respectively). However, those that did complete high school were more likely to report low food security among adults (23.3% of those that completed high school versus 12.7% of those that did not). Therefore, although a 4 × 2 Pearson Chi square analysis revealed a statistically significant difference in categorical food security among adults between those that did and those did not complete high school χ^2^ (3, n = 400) = 7.857, *p* < 0.05, V = 0.140, this effect was not present for binary food security/food insecurity among adults, nor did high school completion affect food security among children. 

With respect to categorical food security at the household level, 67.8% of males reported very low household food security, compared with 55.2% of females; 26.4% of females versus 17.4% of males reported low food security, 4.1% of males and 10.5% of females reported marginal food security, and 10.7% of males versus 7.9% of females reported high food security. A 4 × 2 Pearson chi-square analysis revealed a statistically significant effect χ^2^ (3, n = 398) = 9.838, *p* < 0.05, V = 0.157. Similarly, single participants were more likely than married participants to report very low household food security (61.8% and 42.3%, respectively), but single participants were also more likely to report high food security (9.5% of single participants versus 3.8% of married). Further, married participants were more likely than single participants to report low food security (42.3% and 20.7%, respectively). A 4 × 2 Pearson Chi square analysis revealed a statistically significant effect χ^2^ (3, n = 400) = 14.184, *p* < 0.05, V = 0.188. However, due to the nature of the differences between categories, when household food security was conceptualized as binary food secure/food insecure, there were no significant differences between married and single participants, nor males and females. 

When food security among children was conceptualized as binary food secure/food insecure, 45.7% of those that were solely welfare-dependent, compared with 26.9% of those that had some form or wage or salary based income, reported food security among children. Results of a 2 × 2 Pearson chi-square analysis revealed a statistically significant effect χ^2^ (1, n = 216) = 4.292, *p* < 0.05. The corresponding Phi value was estimated at 0.141, indicative of a small effect according to Cohen’s [35] guidelines. The significant difference between those that were solely welfare-dependent and those that had some form of wage or salary based income was not present when food insecurity was disaggregated into the three categories (very low, low, and marginal or high).

Appendix A present the results of the Chi-squared analyses, with significant results indicated with bold text.

### 4.4. Food Security and Household Composition

Table 3 details the results of the Spearman correlation analyses and the associated descriptive statistics. As can be seen in Table 3, there was a strong, positive correlation between household size and mean person nights. The relationship between household size and adult food security was not significantly correlated, however, there was a significant, negative correlation between mean person nights and adult food security. Although this effect was small, based on Cohen’s [35] guidelines, it did imply that as the mean number of people staying with the participant increased, food insecurity decreased.

### 4.5. Food Security and Food-Related Service Use

The distribution of food-related service use was not sufficiently normal (skew = 2.10, kurtosis = 5.94), and thus a Spearman correlation was conducted between adult food security and food-related service use (M = 1.95; SD = 2.17). There was a medium, positive correlation between adult FSSM scores and food-related service use, based on Cohen’s [35] guidelines (rs = 0.36, *p* < 0.001). As food-related service use increased, food insecurity increased.

### 4.6. Food Security, Mental Health and Quality of Life

The descriptive statistics and associated Pearson correlations for the adult FSSM scores, WHO 5, WHOQOL-BREF scores, DASS-21 scores and the total number of chronic health conditions are presented in Table 4. As can be seen in Table 4, all scores were significantly correlated with each other. Wellbeing (WHO-5) and adult food security yielded a negative correlation, although this was a small effect, based on Cohen’s [35] guidelines. As food insecurity increased, wellbeing decreased. Similar correlations were observed between adult food security and the WHOQOL-BREF physical health and social relationships domains. Adult food security and the WHOQOL-BREF environment and psychological domains were negatively correlated; this was a medium effect, based on Cohen’s [35] guidelines. As food insecurity increased, quality of life across all domains decreased, particularly the environment and psychological domains. Additionally, adult food security was positively correlated with the DASS21 depression, anxiety and stress scores (all medium effects), meaning that as food insecurity increased, depression, anxiety and stress increased. Small positive correlations were also observed between adult food security and loneliness and the number of chronic health conditions. As food insecurity increased, loneliness and the number of chronic physical health conditions increased.

## 5. Discussion

This paper has examined the prevalence and correlates of food insecurity within a cohort of socioeconomically disadvantaged Australian service users. Using the multi-item FSSM, we find that the rate of food insecurity is markedly higher among this cohort than among the general population, for which the rate of food insecurity, albeit measured with a single item, has been consistently estimated to be approximately 5% [3,11,12]. The rate of food insecurity in this cohort (80.8% among adults and 58.3% among children) is extremely high and, in light of the relationships between food insecurity and decreased economic, social, and health outcomes [13,15,22,23], carries with it significant public health and public policy implications.

Despite low income being the strongest and most consistent predictor of food insecurity in the international literature, previous studies have found that demographic characteristics such as age, marital status, household composition, and type of housing tenure have effects on food insecurity independent of income [11,23,36]. This was not the case for our sample; no significant differences in food insecurity were found between those of different sexes, cultural identification, education levels, household compositions, labor force participation status, nor income source. A small significant effect was present for food insecurity among children for those that were welfare dependent, such that those that were welfare dependent reported higher food security than those that had some form of wage or salary based income. 

Given that the sample of this study comprised service users identified as experiencing entrenched disadvantage there are a couple of potential explanations for the lack of effect of socio-demographic characteristics on food security. It may be that there is a very low-income threshold, under which, a person is very unlikely to experience food security. Prior examinations of food insecurity and demographic characteristics in Australia have examined a particular income segment, such as low to middle income earners [11] or have dealt with population representative data and thus a representative distribution of income [23,36]. As average income in Australia is quite high, population-representative studies will not provide accurate insight into the prevalence and correlates of food insecurity in the very low-income margins. Moreover, low income in itself is neither a necessary nor sufficient condition for entrenched disadvantage, though it is clearly correlated with hardship and disadvantage. A family may be in a low-income position but may have assets or have strong family and community support networks, decreasing their need relative to a family in a low-income position without these supports. 

Another potential explanation for the lack of variance in levels of food insecurity between different socio-demographic groups is that the sample were those accessing non-government services, which may well include food emergency relief. Thus, food insecurity may be a driver of service use that precipitated recruitment into the study. This is supported by the result that food insecurity was positively correlated with the number of food emergency relief services the participant had accessed in the 12 months prior to the survey. This result indicates that people with low food security are accessing a greater number of services in order to acquire food. Booth and Whelan [37] comment on the massive expansion of the food banking industry in Australia. They note that, though it is essential that those that are hungry are provided with food, and the provision of food often serves as an entry into other, much-needed services, the stated goal of the food banking industry is to address (immediate) hunger, rather than to address the issues that create demand for food relief. This suggests that some of the participants in this study are relying on food-related services to address their food insecurity, and this explains why these supports are not effectively reducing that food insecurity and may in fact be maintaining their food insecurity.

In examining the impact of household size, firstly, the traditional method whereby participants are asked to report the number of people that usually stay in their dwelling was used. Secondly, a new method was employed to determine the household reference point, whereby the concept of “mean person nights” was introduced. Participants were asked to report the average number of nights per week that people they identified as being part of their family (that is, those that rely on each other for day to day living and may share resources and/or social support) stayed with them. This novel method sought to address some of the limitations of measuring solely a count of usual household members. The number of people who usually stay in the dwelling does not account for cases where participants are sharing with roommates who do not regularly share resources. In these cases, individuals within the same household may have different levels of food security and, therefore, it is not meaningful to compare the food security of one individual in a share house with the food security of individual in a household of the same size, but within a family. Additionally, participants may have custody arrangements where their children regularly stay with their other parent. In these cases, participants may or may not include these children in the number of people that usually stay in their dwelling. There are also cases where other family members, who are members of other households, may regularly stay at the dwelling of the participant, which would not be included in the number of people that usually stay in their dwelling. 

Although there was a strong, positive correlation between mean person nights and household size, only mean person nights and adult food insecurity yielded a statistically significant, negative correlation, albeit a small effect. This negative correlation suggests that larger families have higher food security. This may indicate that larger families pool their resources; indeed, the pooling of resources may be why family members that are not usually in one’s household are temporarily but frequently joining the household. The greater number of people staying in the household may also increase the ability to access emergency relief, including food emergency relief, both in terms of an increased number of people who are ‘eligible’ for emergency relief, and an increased number of people that can visit services, which are often spread throughout the metropolitan area. The statistical significance of the correlation between mean person nights and adult food security suggests that mean person nights is a more accurate measure of the number of people supported by a family unit. It is not surprising that the correlation was weak, as single person families comprised a large proportion of the sample, so it is likely that there is a great deal of variability in food security within single person families. 

Increased food insecurity was associated with reduced scores on the WHO-5 Wellbeing Index, as well as on the physical health, social relationships, environment and psychological domains of the WHOQOL-BREF. Food insecurity was also positively associated with loneliness, depression, anxiety, and stress DASS21 scores, and the number of reported chronic health conditions. Though using a different set of instruments to measure physical and mental health, these findings support several previous studies in Australia and other high-income countries that find that the food insecure are more likely to self-assess their health as poor [18,21,26], more likely to experience stress related to food [9], as well as generally high stress and anxiety levels [3]. There are several theories regarding the relationship between food security and physical and mental health outcomes [13]. It is proposed that the inability to meet nutritional needs leads, in the short-to-medium term, to nutrient deficiencies, metabolic changes, and stress which, in turn, lead to increased prevalence of chronic disease among the food insecure in the long term. An alternative theory is that poor general health, poor mental health, and chronic disease lead to debilitation, decreased economic participation, and lower income, which increases the incidence of food insecurity [13].

Irrespective of the nature of the relationship between food insecurity and physical and mental health, the findings of this study have significant implications for public health practice and policy. We find that access to food emergency relief access is very prevalent, however, access to food emergency relief did not result in higher food security, indicating that food emergency relief is not addressing the underlying causes of food insecurity.

Food insecurity in developed countries is often driven by the increasing cost of food [12] and several practitioners have called for increased study into the monetary and time costs of maintaining a healthy diet in Australia, including variations between States and Territories and between urban and regional/remote areas [37,38,39,40]. There are substantial inequalities in the costs of maintaining an acceptable diet between those facing hardship and more affluent people; relative to households with middle or high income, those facing hardship spend less money but a larger proportion of household budget on food, and spend more time preparing and cooking food, but less time eating [24,39]. Further, as food is the most elastic component of the poor household’s budget, it is often the area that is sacrificed in the face of other expenses, such as debts or unexpected household bills [28]. In order to address food insecurity, national nutrition policy and welfare policy must take these inequalities into account and focus on increasing reliable access to nutritional food, such as through the provision of food vouchers additional to welfare payments and subsidised groceries. Moreover, there is a need to address the underlying structural drivers of entrenched disadvantage and deficiencies in social protection which result in government income support payments below poverty line levels [41,42].

In addition, though our study did not find significant differences between demographic characteristics, past studies in Australia find that being born outside of Australia is negatively related to food insecurity, which may be a result of cooking and budgeting skills being taught and/or prioritized more in other cultures [13,24]. Therefore, there is a role for education in increasing food security in Australia, both in formal education, and through the teaching of these skills in programs designed to alleviate food insecurity.

The impetus for these policy and practice changes is clear. In addition to the human suffering associated with food insecurity, the poor health and mental health outcomes associated with food insecurity affect one’s ability to participate socially and economically which, in turn, results in negative economic consequences, in the form of increased service use and decreased productivity, as well as negative cultural consequences in the form of increased inequality and decreased quality of life.

## 6. Conclusions

The present study examines the prevalence and correlates of food insecurity among a cohort of disadvantaged service users in metropolitan Australia. Using the USDA Household Food Security Survey Module, we find that the prevalence of food insecurity among our cohort greatly exceeds Australian population estimates derived from the single-item measure of food insecurity. Contrary to existing literature, we do not find any significant difference in levels of food insecurity between groups of different demographic characteristics, which may indicate that there is a lower threshold of income, under which food security cannot be achieved, irrespective of other factors that usually enhance food security. These results also reflect that recruitment into the study was via contact with community services and thus the established existence of hardship and need. 

We also find that food insecurity is negatively related to household size, measured by the mean number of ‘person nights’ spent in the household. This may indicate that larger families stay together, outside of their usual households, in order to pool resources. Finally, in line with other studies, we find that food insecurity is related to increased depression, stress, and anxiety, poorer wellbeing and quality of life, and increased incidence of chronic health conditions. Though not without limitations, for instance, the use of self-report measures and the measurement of correlation which does not indicate the direction of relationships between variables, the study reveals a strong need for increased access and stability of access to adequate quantities and quality of food among low-income Australians.

## Figures and Tables

**Table 1 ijerph-17-00559-t001:** Key sociodemographic characteristics of the sample.

	Male	Female	Total *
n (%)	121 (30.3%)	277 (69.3%)	400 (100.0%)
Mean age (years)	46.2	43.0	43.9
Aboriginal and Torres Strait Islander, n (%)	34 (28.1%)	99 (35.7%)	133 (33.3%)
Australian-born, n (%)	97 (80.2%)	213 (76.9%)	312 (78.0%)
Permanent physical disability (self), n (%)	36 (29.8%)	46 (16.6%)	82 (20.5%)
Employed, n (%)	14 (11.6%)	38 (13.7%)	52 (13.0%)
Household composition			
- Single adult	58 (47.9%)	50 (18.1%)	108 (27.0%)
- Two or more adults, no children	29 (24.0%)	47 (17.0%)	76 (19.0%)
- Single adult with child(ren)	8 (6.6%)	97 (35.0%)	105 (26.3%)
- Two or more adults with child(ren)	20 (16.5%)	79 (28.5%)	99 (24.8%)
Accommodation circumstances the night before survey			
- Homeless **	40 (33.0%)	27 (9.7%)	69 (17.3%)
- Public/community housing	44 (36.4%)	122 (44.0%)	166 (41.5%)
- Private rental	28 (23.1%)	99 (35.7%)	127 (31.8%)
- Own house (purchased or mortgaged)	9 (7.4%)	29 (10.5%)	38 (9.5%)

***** Total includes participants that did not identify as binary male or female. Data for non-binary participants are not presented separately as n ≤ 5. ** Includes sleeping rough, staying with friends and family due to having nowhere else to stay, short–medium term accommodation for the homeless, and temporary accommodation.

**Table 2 ijerph-17-00559-t002:** Prevalence of food insecurity—proportion of the sample in each category of food security on the USDA Food Security Survey Module.

	Household (Family Unit)	Among Adults	Among Children
Very low food security, n (%)	237 (59.3%)	248 (62.0%)	24 (11.1%)
Low food security, n (%)	94 (23.5%)	75 (18.8%)	102 (47.2%)
Marginal food security, n (%)	34 (8.5%)	36 (9.0%)	90 (41.7%) *
High food security, n (%)	35 (8.8%)	41 (10.3%)	

* The Child Food Scale comprises three categories, with high and marginal food security considered one category.

**Table 3 ijerph-17-00559-t003:** Spearman correlations and associated descriptive statistics for mean person nights, household composition, and food security scores.

	1.	2. ^†^	M	SD	Skewness	Kurtosis
1. Mean person family unit nights in the household	1.0		2.55	1.90	2.19	7.98
2. Household size (based on usual residence) ^†^	0.76 ***	1.0	3.00	2.41	3.05	16.35
3. Adult Food Security Survey Module (FSSM)	−0.15 **	−0.10	5.98	3.35	−0.43	−1.09

^†^ n = 396, as four participants did not answer this question. ** *p* < 0.01; *** *p* < 0.001.

**Table 4 ijerph-17-00559-t004:** Pearson correlations and associated descriptive statistics for World Health Organization (WHO)-5 Wellbeing Index 5, WHO Quality of Life—Brief (WHOQOL), DASS-21 and chronic health conditions.

	1.	2.	3.	4.	5.	6.	7.	8.	9.	10.	M	SD	Skewness	Kurtosis
1. Adult FSSM	1										5.98	3.35	−0.43	−1.09
2. WHO 5	−0.16 **	1									12.63	6.50	0.07	−0.95
3. WHOQOL Physical Health	−0.17 **	0.61 ***	1								12.72	3.30	−0.13	−0.67
4. WHOQOL Social Relationships	−0.14 **	0.40 ***	0.39 ***	1							12.56	3.76	−0.41	−0.36
5. WHOQOL Environment	−0.31 ***	0.51 ***	0.54 ***	0.44 ***	1						12.66	2.76	−0.07	−0.48
6. WHOQOL Psychological	−0.21 ***	0.67 ***	0.59 ***	0.53 ***	0.59 ***	1					13.00	3.15	−0.31	−0.40
7. DASS21 Depression	0.22 ***	−0.64 ***	−0.55 ***	−0.46 ***	−0.51 ***	−0.70 ***	1				6.55	5.03	0.69	0.00
8. DASS21 Anxiety	0.28 ***	−0.45 ***	−0.49 ***	−0.34 ***	−0.41 ***	−0.47 ***	0.72 ***	1			5.44	4.45	0.87	0.57
9. DASS21 Stress	0.24 ***	−0.57 ***	−0.49 ***	−0.30 ***	−0.43 ***	−0.55 ***	0.76 ***	0.77 ***	1		7.36	4.71	0.41	−0.31
10. Three-Item Loneliness Scale	0.19 ***	−0.49 ***	−0.42 ***	−0.48 ***	−0.38 ***	−0.52 ***	0.58 ***	0.47 ***	0.50 ***	1	6.02	2.04	0.04	−1.17
11. Number of Chronic Health Conditions	0.15 **	−0.22 ***	−0.50 ***	−0.13 **	−0.24 ***	−0.19 ***	0.29 ***	0.35 ***	0.29 ***	0.17 ***	3.48	2.94	0.91	0.34

** *p* < 0.01; *** *p* < 0.001.

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
