# Peer review of "Food Insecurity and Socioeconomic Disadvantage in Australia"

_ijerph, 2020, doi:10.3390/ijerph17020559_

Round 1

Reviewer 1 Report

This study analyses the predictors associated with higher food insecurity within a low-income sample. In addition, it explores the relationship between service use and food insecurity as well as the impacts of food insecurity on health, mental health and quality of life. The comments are listed as follows:

1. IJERPH is an international journal. The authors should elaborate more on the international situation of food insecurity in order to engage with the wider readership of the journal. What is the difference between Australia and other countries? You should present clearly and this can help the paper to attract more international readers.

Furthermore, the cohort of people investigated in this study is of low-income status and they use the non-government service. What does the literature show about this cohort of people?  

2. Methods- Results: While the results are presented well, there is a need for more advanced statistical analysis like the econometrics. The results might be more interesting if the authors decide to use a regression model. I recommend the authors to check “Introductory Econometrics” by Wooldridge for the first point, and then “Regression models for Categorical and Limited Dependent Variables” by Long and Freese.

Furthermore, please check the subsection 4.2 Prevalence of food insecurity. In lines 207-208 you might want to say that the sample experiences very low food security…

3. Discussion. The authors constantly compare their results with the general population of Australia (e.g. line 280). What about other studies that have used cohorts of similar socioeconomic status with this study? Are results from this study in line with others from previous research?

4. Please correct the numbering in references as well as in line 152 the punctuation mark (double full stop).

Author Response

IJERPH is an international journal. The authors should elaborate more on the international situation of food insecurity in order to engage with the wider readership of the journal. What is the difference between Australia and other countries? You should present clearly and this can help the paper to attract more international readers.

Furthermore, the cohort of people investigated in this study is of low-income status and they use the non-government service. What does the literature show about this cohort of people?  

Firstly, thank you very much for your detailed and considered review of our paper.

We feel that we reference the food insecurity situation in the international context quite extensively e.g. p1 paragraph 1 (lines 27-35) sets the context as high-income countries (not just Australia; p1 paragraph 2 (lines 36-45) outline the measurement and prevalence of food insecurity in Canada, NZ, and the US. The remaining paragraphs refer to themes found in the international literature and then relate them to findings found in Australia, as this is the geographic context of our study – e.g. p2 line 56 – this paragraph outlines that, in high income contexts, food insecurity varies among substrata of the populations, then presents Australian-specific figures.

With respect to the cohort of this study, p2 lines 65-92 outline the factors that affect food insecurity. Fundamentally, low income is the strongest predictor of food insecurity therefore the literature shows that this cohort experiences greater food insecurity. There is not, to our knowledge, a strong body of literature on food insecurity and non-government service use. There are several articles about food insecurity and health service usage, but these are not relevant to our context.  

Methods- Results: While the results are presented well, there is a need for more advanced statistical analysis like the econometrics. The results might be more interesting if the authors decide to use a regression model. I recommend the authors to check “Introductory Econometrics” by Wooldridge for the first point, and then “Regression models for Categorical and Limited Dependent Variables” by Long and Freese.

Furthermore, please check the subsection 4.2 Prevalence of food insecurity. In lines 207-208 you might want to say that the sample experiences very low food security…

Thank you for picking up the error on line 207/8, this has now been corrected. We respectfully disagree that econometric modelling is indicated here, as we contend that food insecurity is not an economic phenomena (p2, lines 56-64). In addition, the correlates we examine are not particularly relevant to econometric models.

We do acknowledge that this feedback may be less about the discipline and more about the arguably less advanced statistical modelling, hence the suggestion for regression modelling. In this regard, we intentionally chose correlations due to competing theories about the relationship between food insecurity and poor outcomes (i.e. does inability to meet nutritional needs lead to nutrient deficiencies, metabolic changes and stress, which in turn leads to chronic diseases such as cardiovascular disease, diabetes and depression? Or does poor general health and chronic disease lead to debilitation, lower economic participation and lower income, which leads to food insecurity? Ramsey, Giskes, Turrell and Gallegos, 2011). These competing theories in the international literature, combined with the comparatively few studies that examine multidimensional food insecurity in Australia, mean that predictive modelling such as regression would not be theoretically robust.

Reference: Ramsey R, Giskes K, Turrell G, et al. Food insecurity among adults residing in disadvantaged urban areas: potential health and dietary consequences. Public Health Nutr. 2012 Feb;15(2):227-37.

Discussion. The authors constantly compare their results with the general population of Australia (e.g. line 280). What about other studies that have used cohorts of similar socioeconomic status with this study? Are results from this study in line with others from previous research?

Thank you for this suggestion. We agree that our discussion does refer to Australia quite a lot and that it is not clear when findings are found internationally. To this end, we have added clarification of where the discussion is referring to findings present in the international literature/literature about food insecurity in developed nations (line 308, 379, and 393). Lines 307-342 refer to the implications of the nature of the cohort on the results, relating the results to extant literature.

Please correct the numbering in references as well as in line 152 the punctuation mark (double full stop).

Thank you for noting these errors– they are now corrected.

Reviewer 2 Report

The article is based on well prepared and executed research. The analysis and calculations were well executed, utilizing the relevant statistical methodology. The conclusions are interesting for the reader and well formulated. The author(s) utilized a broad selection of sources published on this topic worldwide. 

Overall the article is an interesting read, it fits well with the profile of the publication and it is my recommendation it should be published.  

Author Response

Thank you for your review!

Reviewer 3 Report

Reviewer’s Comments.

Thanks for the opportunity to review this paper.  I enjoyed reading the paper. I think this is a well-researched work. I commend authors for providing adequate background literature on the subject. The goals and rationale of the study are clearly stated in the paper. Also, measures used to collect data and data collection procedure and analysis are well described.  The presentation of the results is good, and discussions of results is quite robust.  

I suggest that authors consider including a table to show the Pearson Chi-square analysis for the differences in food security by the different demographic characteristics (please see page 5-6). This may help readers to better understand how those characteristics affect food security.  Also, on table 2, page 5, why did authors put marginal and high food security percentages and samples together for the children category? Authors should explain why they did this in the text.

Author Response

Thank you very much for your helpful comments!

We really appreciate the suggestion to include a table of the Chi-square test results, however, because so many variables were tested (sex, Indigeneity, marital status, household composition, educational attainment, labour force status, and welfare dependence) against three different types of food security (household, adult, and child), the presentation is very inelegant and, as the vast majority results are non-significant, we feel would detract from the key points of our paper. We do acknowledge that the results may be helpful to some readers and have therefore prepared the tables suggested for inclusion as supplementary materials (Tables S1-S6), made reference to these tables and expanded on the results in-text.

With respect to the marginal and high food security among children, these are not combined percentages, this the label of the food security among children category on the USDA FSSM. This is outlined in the measures (p4, line 152), but we agree that it may be confusing/misleading in the results so we’ve added the following note on p5, line 209 to clarify this “(note that the USDA FSSM labels high or marginal food security among children as a single category)”.
